# Homogeneous versus Inhomogeneous Polarization Switching in PZT Thin Films: Impact of the Structural Quality and Correlation to the Negative Capacitance Effect

**DOI:** 10.3390/nano11082124

**Published:** 2021-08-20

**Authors:** Lucian Pintilie, Georgia Andra Boni, Cristina Florentina Chirila, Viorica Stancu, Lucian Trupina, Cosmin Marian Istrate, Cristian Radu, Ioana Pintilie

**Affiliations:** National Institute of Materials Physics, Atomistilor 405A, 077125 Magurele, Romania; andra.boni@infim.ro (G.A.B.); dragoi@infim.ro (C.F.C.); stancu@infim.ro (V.S.); Lucian.Trupina@infim.ro (L.T.); cosmin.istrate@infim.ro (C.M.I.); cristian.radu@infim.ro (C.R.); ioana@infim.ro (I.P.)

**Keywords:** polarization switching, epitaxial, ferroelectric, thin films

## Abstract

Polarization switching in ferroelectric films is exploited in many applications, such as non-volatile memories and negative capacitance field affect transistors. This can be inhomogeneous or homogeneous, depending on if ferroelectric domains are forming or not during the switching process. The relation between the polarization switching, the structural quality of the films and the negative capacitance was not studied in depth. Here, Pb(Zr_0.2_Ti_0.8_)O_3_ (PZT) layers were deposited by pulse laser deposition (PLD) and sol-gel (SG) on single crystal SrTiO_3_ (STO) and Si substrates, respectively. The structural quality was analyzed by X-ray diffraction and transmission electron microscopy, while the electric properties were investigated by performing hysteresis, dynamic dielectric measurements, and piezo-electric force microscopy analysis. It was found that the PZT layers grown by PLD on SRO/STO substrates are epitaxial while the layers deposited by SG on Pt/Si are polycrystalline. The polarization value decreases as the structure changes from epitaxial to polycrystalline, as well as the magnitude of the leakage current and of the differential negative capacitance, while the switching changes from homogeneous to inhomogeneous. The results are explained by the compensation rate of the depolarization field during the switching process, which is much faster in epitaxial films than in polycrystalline ones.

## 1. Introduction

Polarization switching in ferroelectrics is exploited in applications as non-volatile memories or memristors [1,2]. Many models were developed to simulate the kinetic of the switching, the most popular being Kolmogorov–Avrami–Ishibashi (KAI) and nucleation limited switching (NLS, also known as the non-KAI model) [3,4,5,6]. These models are based on the assumption that, starting from a certain orientation of polarization, if an electric field of opposite direction is applied on the sample, then nuclei of polarization with orientation parallel with the applied field are formed first, followed by their growth until the polarization changes orientation in the entire volume of the sample. It is assumed that the nuclei form on structural defects that can induce local strain and internal electric fields favoring the new orientation of polarization. This type of switching is also known as being inhomogeneous, because it does not take place in the entire volume at the same time. Opposite to this is the homogeneous switching, in which polarization direction suddenly changes, without domain formation [7]. Such switching is very fast and it was suggested that it can be induced by irradiation with infrared or THz light of suitable intensity [8,9]. Other models were also developed, considering various intrinsic and extrinsic influences on the switching and trying to simulate the polarization-electric field (P-E) hysteresis loops, and various experiments were imagined to test these models [10,11,12,13,14,15].

Related to polarization switching is the negative capacitance (NC) effect, reported in the ferroelectric capacitors and in the ferroelectric based multilayers [16,17,18,19,20,21]. It is thought to be a solution to overcome the Boltzmann tyranny, meaning to reduce the slope of the sub-threshold swing in ferroelectric field effect transistors (FET) below the predicted limit of 60 mV/decade, although recent studies show that the enhancement is questionable [22,23,24]. In some reports, NC was evidenced as a transitory effect during polarization switching; in others, it was claimed to be a stationary effect in specially designed super-lattices combining ferroelectric and dielectric layers. In any case, it is largely accepted that NC is present when ferroelectric polarization is close to a zero value, such as around the coercive electric field during switching, around ferroelectric–paraelectric phase transition, or in the center of polarization vortices [25,26,27,28,29,30,31,32]. The impact of ferroelectric domains on the presence and magnitude of the NC effect is also a matter of debate, as there are studies reporting that the presence of domains is detrimental for NC while other studies claim that the presence of ferroelectric domains is beneficial for NC [33,34,35].

Although there is an abundant literature on both subjects, polarization switching and NC effect, the relation between these phenomena and the sample’s structural and electrical properties is less studied. One can hardly find a comprehensive study on this matter that mentions the impact of homogeneous or inhomogeneous switching on the NC effect. Here, it is shown that the polarization switching is changing from homogenous to inhomogeneous as the structural quality of the ferroelectric films changes from high quality epitaxial to polycrystalline, and the leakage current drops with about four orders of magnitude. The changes in the switching kinetic are accompanied by marked changes in the magnitude of the NC effect, which is directly affected by the structural quality, the domain structure and the magnitude of the leakage current. It is also an experimental confirmation that the presence of ferroelectric domains is detrimental for the presence of NC, a finding that has to be considered when designing gate structures for NC-FET.

## 2. Materials and Methods

The study is performed on epitaxial and polycrystalline Pb(Zr_0.2_Ti_0.8_)O_2_ (PZT) ferroelectric films in capacitor geometry. It is found that the magnitude of the NC effect is drastically reduced as the structural quality degrades from epitaxial to polycrystalline and when the magnitude of the leakage current is reduced. This is attributed to domain formation due to incomplete or slow screening of the depolarization field, leading to back-switching and elongated hysteresis loops. In principle, lower leakage and domain formation leads to a lower magnitude of NC effect, but may extend the time duration; thus, by controlling the structure and the leakage current one can control the NC in FET devices.

Epitaxial PZT films were grown by pulsed laser deposition (PLD) on single crystalline SrTiO_3_ (STO) substrates with (001) orientation having dimensions of 10 × 5 × 0.5 mm^3^, after previous deposition of a 20 nm thick SrRuO_3_ (SRO) bottom electrode. The growth of the films was performed by pulse laser deposition (PLD) from targets with different purities, considering that the magnitude of the leakage current can be influenced by the amount of impurities acting as donors or acceptors. For PLD deposition a workstation from Surface GmbH was used, with a KrF excimer laser having a 248 nm wavelength. One of the targets has 99.9% purity and was purchased from a target manufacturer (will be named as a commercial target), while the other target has 99.99% purity and was in-house produced using high purity precursor oxides (PbO 99.999%; TiO_2_ 99.99%; ZrO_2_ 99.99%, all from Puratronic (Sigma-Aldrich, Saint-Louis, MO, USA); will be named a pure target) [36]. The concentration of the impurities in the targets was estimated by performing inductive coupled plasma-mass spectroscopy (ICP-MS) and X-ray fluorescence experiments. It was found that the number of impurities acting as donors or acceptors is at least two times larger in the commercial target compared to the pure one [36]. Polycrystalline PZT films were deposited by spin-coating on SRO/STO single crystal substrates (SRO deposited by PLD) and on Pt-coated Si substrates. Details about target preparation, compositional analysis and deposition parameters can be found in the Appendix A.

The structural quality was investigated using X-ray diffraction (XRD) and high-resolution transmission electron microscopy (HR-TEM, using JEM-ARM 200 F from JEOL, Tokyo, Japan). XRD investigations are performed using Rigaku SmartLab, Japan, having an X-ray source with a Cu anode, powered at 40 kV and 40 mA, in a parallel beam, with a Ge (220) monochromator in the incident beam and using a Hypix detector in the 0D regime; for the reciprocal space mappings (RSM), a Bruker D8 Advance diffractometer (Billerica, MA, USA) was used, in parallel beam, without a monochromator. Ferroelectric properties at microscale were investigated using piezo-force microscopy (AFM - MFP-3D-SA with PFM facility from Asylum Research/Oxford Instruments, Oxford, UK), using a specially designed scanning method to reveal the polarization switching mechanism, with or without domain formation.

For the electrical measurements, top electrodes are deposited with an area of 0.01 mm^2^. The thickness of each layer is evaluated from TEM images. For PLD grown layers, the thickness is about 160–170 nm, while for sol-gel deposited films the thickness is in the 250–300 nm range. The dynamic dielectric characterization [37] was used to extract the dependence of the film resistance R_p_, the film capacitance C_p_ and the capacitance of the structure during switching C_total_ on the amplitude of the trapezoidal voltage used in measurements (trapezoidal pulses of same time duration and of increasing amplitude are applied on the sample and the corresponding current is recorded; from the current magnitude at different times elapsed from the starting of the voltage pulse one can estimate the above mentioned quantities, see also SI for details). Standard hysteresis loops, capacitance-voltage (C-V) and current-voltage (I-V) characteristics were also recorded at room temperature. These were used to extract information regarding the magnitude of the leakage current, dielectric constant and remnant polarization. These measurements were performed in a Lake Shore cryo-station (Lake Shore Cryotronics, Westerville, OH, USA) with micromanipulated arms and CuBe needles for contacting the electrodes. A TF2000 ferristester from Aix ACCT (AixACCT Systems GmbH, Aachen, Germany) was used for the dynamic dielectric characterization and for recording polarization–voltage hysteresis loops. I–V characteristics were recorded using a Keithley 6517B electrometer (Tektronix, Beaverton, OR, USA) with an incorporated dc voltage source.

## 3. Results

### 3.1. Structural Quality

The low magnification TEM image from Figure 1a displays in diffraction contrast the multilayered PZT/SRO/STO sample. The PZT layer has a thickness of, approximately, 160 nm. The electron diffraction pattern from Figure 1b corresponds to a selected area including the substrate and the layers, as is suggested by the TEM image at low magnification from Figure 1a. The selected area electron diffraction (SAED) pattern in Figure 1b reveals the crystallization status of the as-deposited layers and the orientation relationship with respect to the STO substrate. The diffraction spots were marked with the corresponding Miller indices and subscripts for the STO substrate and the PZT thin film. The diffraction pattern reveals the epitaxial growth of the SRO and PZT layers. From the measurements performed on the SAED figure, there is a clear orientation relationship between the crystallographic planes in the STO substrate and the PZT thin film, namely: (001)_PZT_ || (001)_STO_ and (010)_PZT_ || (010)_STO_. The mismatch between (010)_PZT_ (d_010_ = 0.3953 nm) and (010)_STO_ (d_010_ = 0.3899 nm) planes, calculated with respect to the STO substrate, is of 1.38%, leading to the overlapping of 010_STO_ and 010_PZT_ diffraction spots in Figure 1b. We can also notice that in the diffraction pattern there are some diffraction spots which appear as if doublets, thus revealing the epitaxial relationship between STO, SRO and PZT layers, as well as the lattice mismatch between the PZT and SRO/STO lattices. One has to mention that the mismatch between STO and SRO lattice constants (both are with cubic perovskite structure) is around 1%, thus the STO and SRO diffraction spots are overlapped in the SAED pattern. Therefore, the SRO subscripts were omitted from indexing the STO diffraction spots.

The high resolution TEM (HRTEM) image from Figure 1c confirms the single crystal growth of the PZT and SRO layers and the well-defined orientation relationship (001)_PZT_ || (001)_STO_ and (010)_PZT_ || (010)_STO_ with respect to the STO(001) substrate. Strain contrast may be observed in the HRTEM images as a consequence of the mismatch dislocations. The strain contrast is also visible on the low-magnification image in Figure 1a, especially along the SRO-PZT interface.

In the case of the polycrystalline sample deposited on the Pt/Si substrate, due to the large difference between the atomic numbers Z in PZT/Pt layers and Si substrate, the TEM specimen may not show transparent areas simultaneously for all materials. In this case, it is possible to have, in some areas, an amorphous or a very thin Si substrate (due to the influence of the ion beam milling), and in other areas to have a thick layer of Pt or PZT.

The TEM image at a low magnification, from Figure 2a, displays in diffraction contrast the multilayered PZT/Pt/Si structure. A bright thin layer (4 nm) can be noticed between the Si substrate and the Pt layer. As was mentioned above, the TEM image at low magnification also reveals a very thin Si substrate, whilst the Pt layer still remains thick. The measured thickness for the PZT film is around 245 nm, also determined from the TEM image at low magnification. The SAED pattern from Figure 2b reveals the crystallization status of the as-deposited layers (PZT/Pt). We can observe a multitude of diffraction spots, from which some were indexed with the corresponding Miller indices and subscripts, indicating the Si substrate and the PZT thin film. It is very clear, from the diffraction pattern, that there is no preferential orientation nor an orientation relationship between the Pt layer and the Si substrate or between the PZT layer and the Pt layer. From the distribution of the spots, the sample is rather polycrystalline with no ordered single-crystal-like growth. XRD analysis has confirmed the TEM results.

### 3.2. Hysteresis and Derivative

Hysteresis measurements were performed at room temperature (RT) using a triangular voltage wave of 1 kHz frequency and increasing amplitude. The polarization-voltage loops were recorded as the amplitude of the voltage wave was increased from zero to maximum value to saturate the ferroelectric polarization. The loops were then graphically derived in order to obtain the voltage dependence of the differential capacitance The obtained results are presented in Figure 3 for the epitaxial PZT layers, and in Figure 4 for the polycrystalline films, respectively.

The magnitude of the NC effect (the maximum value of the negative capacitance) was highest in the epitaxial film deposited from the commercial target (see Figure 3c,e). In the film deposited from the pure target, it was about 30% lower (see Figure 3d,f), while in the PZT layer deposited by sol-gel on SRO/STO substrate it was already four to five time lower than in the film deposited from the commercial target (see Figure 4c,e). The lowest value, of about 30 times lower than in the case of the PZT epitaxial film deposited from commercial target, was obtained in the case of the polycrystalline film deposited on Pt/Si substrate (see Figure 4d,f).

One can also observe that the hysteresis loops for the epitaxial films are almost rectangular and with sharp switching. The hysteresis loop retains a rectangular shape in the case of the polycrystalline film deposited on the SRO/STO substrate, but with a significantly lower value of remnant polarization. For the film deposited on Pt/Si, the loop is more elongated, with an even lower value of the remnant polarization and with a larger coercive field (see Table 1). These differences will be analyzed in more detail later on in a discussion section.

Analyzing the values in Table 1, one can see that the value of the remnant polarization decreases as the structural quality changes from highly epitaxial to polycrystalline. This is because all the structural defects can affect the polarization switching, pinning the domains and restricting the movement of the domain walls. On the other hand, the coercive file does not vary too much, suggesting that the change of the orientation of the electrical dipole in each unit cell takes place at about the same applied electric field no matter the structural quality. What is changing is the amount of volume in which the polarization can be switched, with impacts on the value of the remnant polarization. Noticeable is the fact that the polarity of the internal electric field is changing from positive (oriented from the bottom electrode towards the top one) to negative (oriented towards the bottom electrode). This fact has impacts upon the domain structure: the epitaxial films have polarization oriented mostly upwards, while the films deposited by sol-gel method, which are polycrystalline, have mixed polarization (up and down). The results will be further confirmed by PFM investigations.

Analyzing Figure 3e,f and Figure 4e,f one can observe that, in all cases, the maximum negative capacitance occurs at about the same voltage at which the current recorded during the hysteresis measurement has a maximum value [26]. However, the magnitude of this maximum current is also decreasing with almost two orders of magnitude, from the epitaxial film deposited from the commercial target to the polycrystalline one deposited by sol-gel on the Pt/Si substrate.

### 3.3. Dynamic Dielectric Method to Extract Rp, Cp and Ctotal

The dynamic dielectric characterization method [37] was applied on the samples and the obtained results are presented in Figure 5. The method was applied in two cases:Sample prepoled by applying a trapezoidal pulse of maximum amplitude then trapezoidal pulses of increasing amplitude and of opposite polarity were successively applied on the sample and the current response was recorded. Polarization switching takes place in this case. The values of Rp, Cp and Ctotal were then estimated using the procedure presented elsewhere [37] and the obtained results were presented as function of the amplitude of the applied trapezoidal voltage pulse.The sample was prepoled as previously mentioned, and then pulses of the same polarity but of increasing amplitude were applied, and the current response was recorded. In this case, there is no polarization switching. Rp, Cp and Ctotal were again estimated and the obtained results were presented as functions of the amplitude of the applied trapezoidal voltage pulse.

Analyzing Figure 5 one can see that:Ctotal has negative values during polarization switching. The larges values are obtained for the film deposited from the commercial target, and the smallest (more than two orders of magnitude smaller) are obtained for the polycrystalline film. If there is no polarization switching, then Ctotal is positive and relatively constant with the amplitude of the applied trapezoidal voltage.Cp has all the time positive values and presents a butterfly shape for the case when polarization switching takes place. If there is no switching, then Cp is almost constant and of the same value with Ctotal. One has to notice that the dynamic dielectric constant gives very nice results for Cp and Ctotal when applied to epitaxial films. In the case of the polycrystalline film the values are more scattered, and the butterfly shape of Cp, associated with polarization switching, is barely visible.Rp value has a drop if polarization switching takes place. When there is no polarization switching then Rp has the tendency to be relatively constant, with some increase around the coercive voltage, probably due to changes in the potential barriers at the Schottky-like contacts that are present at the electrode interfaces.The results of the dynamic dielectric characterization method are in agreement with those obtained from hysteresis measurements. NC effect is more pronounced in the epitaxial films than in polycrystalline ones. PFM investigations were further performed to see how domain configuration relates to NC.

### 3.4. PFM Results

We obtained a PFM scan with voltage increasing from maximum negative value to maximum positive value.

PFM investigations were performed using the same procedure as described elsewhere [26]. The results are presented in Figure 6 and Figure 7 for epitaxial and polycrystalline films, respectively.

The analysis of the images presented in Figure 6 and Figure 7 (see also Appendix A in Appendix A) reveal the following:For the film obtained from the commercial target the well-known grid of 90° domain walls is obtained [36]. As the dc voltage applied on the tip increases from zero to the maximum value (from the left to the right in the PFM phase and PFM amplitude images), the polarization suddenly changes its orientation from upward (towards the surface, on the left side) to downward (towards the bottom electrode, on the right side). Notably, the grown film has a dominant upward orientation of polarization [38,39]. No mixture of domains with upward and downward directions of polarization is visible during switching. One can presume that the switching time is so fast that it is not possible to visualize this mixture of domains with almost zero value of the total polarization.For the films obtained from the pure target, the situation is significantly different. One can see that the domain structure is much more complicated than in the case of the film deposited from the commercial target. Outside the central square, where the dc voltage was gradually increased from the maximum negative voltage to the maximum positive voltage, there is a mixture of domains with upward and downward polarization directions, in contrast with a dominant upward direction evidenced in the film deposited from the commercial target. Additionally, looking to the central square one can see that the switching is not sharp and that there is a voltage range (a region) in which mixture of domains with upward and downward polarization directions are present. This region is flanked by a region of upward polarization to the left and one of downward polarization to the right. One can also notice that the grid of 90° domain walls becomes visible only after poling the film upward or downward, while on the grown surface outside the central square, this grid is hardly visible.In the case of the polycrystalline film, the switching takes place with formation of mixed domains, as expected. One can also notice that the grown surface is a mixture of domains with opposite directions of polarization [40,41,42,43,44,45,46,47,48].As expected, the polycrystalline films are rougher than the epitaxial ones, with RMS values in between 5 and 15 nm, compared to 0.5 nm for epitaxial ones.

### 3.5. Leakage Current

I-V characteristics at room temperature, recorded for epitaxial PZT films deposited from commercial and pure targets and for the polycrystalline PZT layers deposited by sol-gel, are presented in Figure 8. One can see that the largest leakage current is recorded for the film deposited from the commercial target, while the lowest leakage current is recorded for the polycrystalline sample deposited on the Pt/Si substrate.

The differences in the magnitude of the leakage current for samples of different purities and/or structural quality can be explained as follows:In the case of epitaxial films deposited from targets of different purities, the larger leakage current for the film deposited from the commercial target can be explained by the larger number of hetero-valent impurities found in this target compared to the pure one. These impurities can act as donors and acceptors and can affect the density of the free carriers on one hand and the height of the potential barriers at the electrode interfaces on the other hand, not mentioning the impact they can have on domain wall movement, reflected in the shape of the hysteresis loop [40,41,42,43,44,45,46,47,48]. It is known that acceptor and donor impurities can introduce energy levels in the gap, with impacts upon the position of the Fermi level position and the band alignment at the electrode interfaces [49,50,51,52]. Depending on the position of the Fermi level, the height of the potential barriers at electrodes, for the majority of carriers, can be larger or lower (in between 0.1 and 0.3 eV [36]), with impacts upon the magnitude of the leakage current.In the case of the polycrystalline films, the even lower leakage current can be caused by the grain barriers, acting both as trapping and scattering centers for the charge carriers.

## 4. Discussion

The experimental results show that the NC effect is a transitory effect, strictly related to polarization switching. Its magnitude, in terms of the value of the negative specific capacitance, is dependent on the structural quality of the ferroelectric film and on the magnitude of the leakage current.

Analyzing the experimental results, the following possible explanation emerges: the negative capacitance effect is more pronounced in the case of homogeneous polarization switching, that it is possible in high quality epitaxial monodomain films, and degrades in the case of the inhomogeneous polarization switching that occurs in polycrystalline layers.

Inhomogeneous switching is characterized by nucleation of domains with an opposite direction of polarization, followed by the growth of these domains until the polarization is fully switched and parallel with the applied electric field. There are many experimental and theoretical studies dedicated to this subject [53,54,55,56,57,58,59,60,61,62,63]. However, one has to consider that, immediately after switching, a strong depolarization field is present in the ferroelectric sample. The compensation of this field requires redistribution of the free and trapped charges that allow the existence of the initial monodomain state. It does not matter where these charges are located, in the ferroelectric film or in the electrodes; what matters is that they need some time to redistribute and to move from one electrode interface to the other in order to compensate the depolarization field and to stabilize the monodomain state with an opposite direction of polarization compared to the initial one. This final step of the polarization switching, the compensation of the depolarization field, is not considered in the popular models used to simulate the switching kinetics, as in, for example, the Kolmogorov–Avrami–Ishibashi (KAI) or nucleation limited switching (NLS) models [4,64,65].

Qualitatively, one can assume that the speed of compensation of the depolarization field by redistribution of free and trapped charges depends on some time constants: the time constant of the ferroelectric sample, which is given by the R_FE_C_FE_ product, where R_FE_ and C_FE_ are the resistance and the capacitance of the ferroelectric layer, respectively; the time constant of the measuring circuit; the emission time constant from the traps. As all the samples were measured with the same experimental circuit, one can assume that its influence over switching is the same for all samples. Additionally, the emission time constant from the traps can be considered the same, as it is related to the activation energy of the trapping level (the depth in the gap compared to conduction band minimum or valance ban maximum, depending on where this level is located in the gap). A certain structural defect, apart from substitutional impurities, is introducing specific trapping levels in the gap, having the same emission time constant in all the samples. What can vary from sample to sample is the density of the defect, that can be larger in polycrystalline films than in epitaxial ones. Substitutional impurities are also introducing energy levels in the gap. These can act as shallow donors or acceptors, thus introducing free carriers in the ferroelectric sample, or can act as deep traps that function through temperature and frequency of the applied electric field, and can exchange carriers with the conduction or valance bands or not. In short, the same defects are present in the samples, whatever they are structural or impurities, they have the same emission time constants, what can vary is their density. It is also clear that a large density of defects can have impacts on the density and mobility of the charge carriers, leading to low values of these quantities, thus to larger values of resistance. Charged defects also contribute to the total capacitance of the sample, reflected in large values for the dielectric constant. A large density of ferroelectric domain walls can also bring contributions to the dielectric constant, because they can easily respond even to varying voltages of low amplitudes.

Considering all these aspects, one can conclude that more defects means higher capacitance and resistance of the ferroelectric sample. This can have the following consequences:A larger time constant for the ferroelectric layer in the case of the polycrystalline films compared to the epitaxial ones;A lower leakage current in polycrystalline films compared to epitaxial ones.

The final consequence is that, as the structural quality changes from polycrystalline to epitaxial, the polarization switching is changing from inhomogeneous, with domain formation, to homogeneous, without domain formation (see also Figure 9 for a better understanding of the difference between homogeneous and inhomogeneous switching).

The homogeneous switching was reported in ferroelectric ultra-thin films, in relation to the intrinsic coercive field predicted by the Landau–Ginzburg–Devonshire (LGD) thermodynamic theory [66,67,68,69,70]. Novel techniques to evidence the homogeneous switching were also suggested, such as illumination with electromagnetic radiation in the THz or infrared domain [8,9,15]. However, the experimental results and the related models are not correlating the homogeneous switching to the structural quality of the samples.

Our results show that homogeneous switching is also possible in relatively thick films (100–200 nm), if these are grown in a mono-domain state. When the density of the structural defects is increasing and the density of the charge carriers involved in the compensation of the depolarization field is decreasing, then the film is no longer mono-domain (see PFM images in Figure 7, Appendix A) and the switching becomes inhomogeneous, obeying the well-known nucleation and growth mechanism. This change of polarization switching from homogeneous to inhomogeneous has also impacted upon the magnitude of the NC effect, which becomes considerably lower in the case of polycrystalline films. This can have detrimental effects on the use of this effect to reduce power consumption in FET devices.

## 5. Conclusions

We have shown in this study that homogeneous polarization switching is possible in high quality epitaxial PZT films, due to the large leakage current allowing rapid compensation of the depolarization field established just after polarization switching. As the crystalline quality is changed to polycrystalline and the leakage current decreases, the switching becomes inhomogeneous. This change is accompanied by a significant reduction in the magnitude of the NC effect. All these results show that polarization switching, structural quality, the magnitudes of the leakage current and NC effect are closely related and should be analyzed together. A possible hypothesis is that the natural switching, the intrinsic one, is homogeneous in ferroelectrics, without domain formation, and that the change to inhomogeneous switching takes place due to extrinsic factors as structural defects acting as nucleation centers for new domains or as pinning centers for domain movements. This hypothesis is closer to the popular thermodynamic or microscopic theories, postulating that there are only two possible values (or orientations) of polarization, with no stable intermediate states. Therefore, the possibility to obtain intermediate polarization states from volumes of domains with opposite directions of polarization is unlikely in this case. However, this can depend on the time scale used to monitor the polarization switching. The PFM studies were performed with low scanning rates; thus, the time was probably long enough for a proper compensation of the depolarization field in epitaxial films, precluding in this way the formation of domains with opposite orientations of polarization. The result is that the polarization appears to suddenly switch the orientation from fully upward (towards the top electrode) to fully downwards (towards the bottom electrode) and vice versa. The hysteresis loops were recorded at a shorter time scale. Some domains can form in this case, around the zero value of polarization. This is supported by the fact that the switching is not a vertical line at the coercive field; it has some finite slopes because the charges needed to compensate the depolarization field needs some time to move from one electrode interface to the other once the polarization orientation has changed. Therefore, further studies are needed to elucidate all the correlations between switching time and kinetic, structural quality, compensation charges movement, and the magnitude of the leakage current or of the NC effect. In any case, the results presented in this study show that epitaxial films can be more suitable for precision high tech applications requiring high value of polarization, sharper switching, and a more pronounced NC effect, although the costs related to deposition methods may be slightly larger than for polycrystalline films.

## Figures and Tables

**Figure 1 nanomaterials-11-02124-f001:**
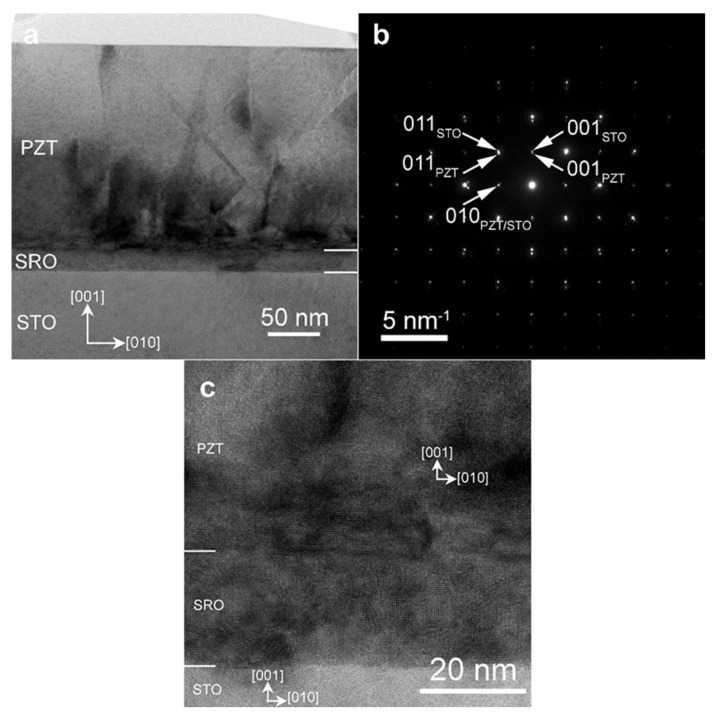
(**a**) TEM image at low magnification of the PZT/SRO/STO structure and (**b**) the corresponding SAED pattern from an area which includes both the substrate and the thin films, (**c**) HRTEM image of the SRO/STO and SRO/PZT interfaces.

**Figure 2 nanomaterials-11-02124-f002:**
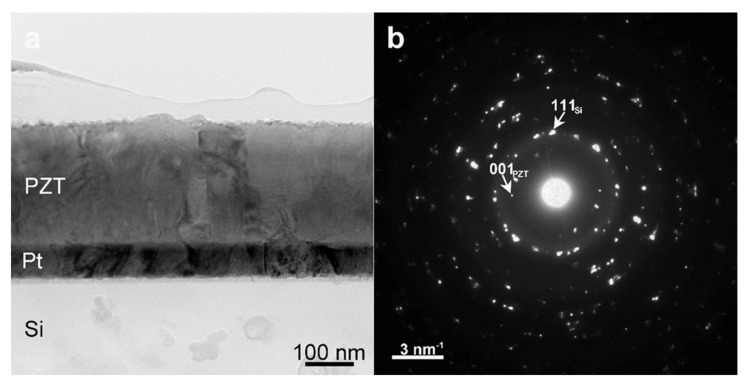
(**a**) TEM image at low magnification of the PZT/Pt/SiOx/Si structure and (**b**) the corresponding SAED pattern performed on area which include both the substrate and the thin films. See also Appendix A in Appendix A for the PZT layer deposited by sol-gel on STO substrate.

**Figure 3 nanomaterials-11-02124-f003:**
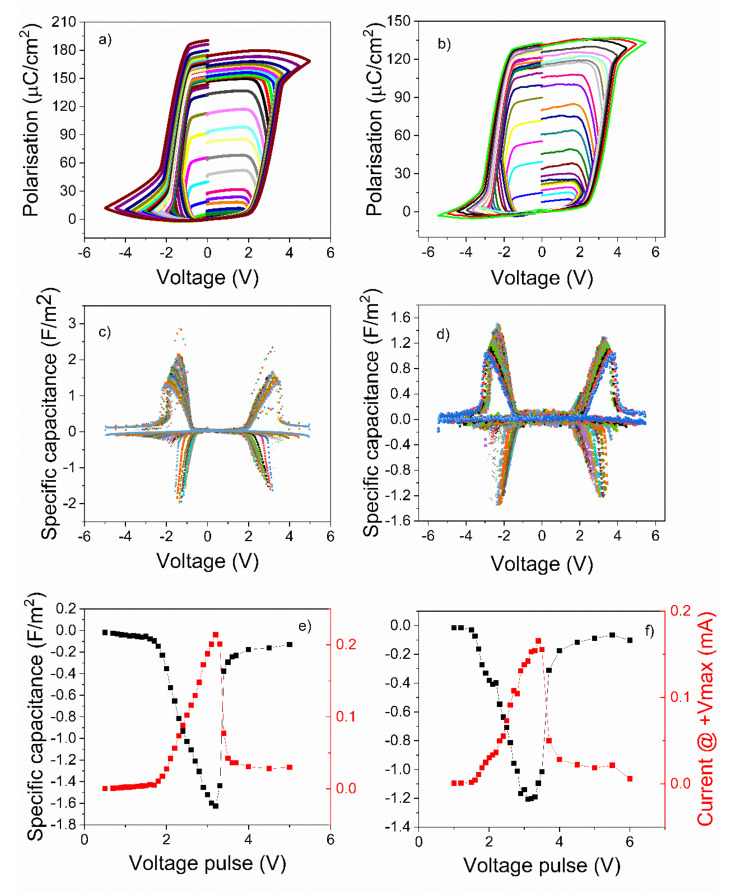
(**a**,**b**) Polarization-voltage characteristics obtained for different amplitudes of the voltage pulse. (**c**,**d**) Specific capacitance obtained from the derivative of polarization-voltage loops. (**e**,**f**) Representation of the maximum values of the specific capacitance and of the current at maximum voltage as a function of amplitude of the voltage pulses. The results from the first column are attributed to the epitaxial PZT thin films deposited from commercial target and the results from the second column are attributed to the epitaxial PZT thin films deposited from “pure” target, respectively.

**Figure 4 nanomaterials-11-02124-f004:**
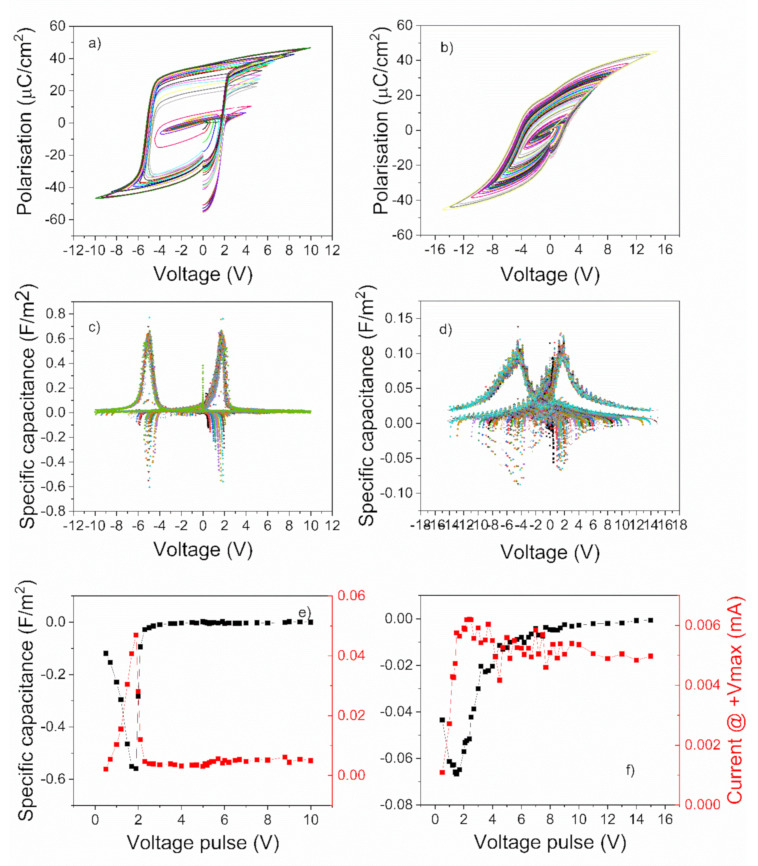
(**a**,**b**) Polarization-voltage characteristics obtained for different amplitudes of the voltage pulse. (**c**,**d**) Specific capacitance obtained from the derivative of polarization-voltage loops. (**e**,**f**) Representation of the maximum values of the specific capacitance and of the current at maximum voltage as a function of amplitude of the voltage pulses. The results from the first column are attributed to polycrystalline PZT film deposited by spin-coating on the SRO/STO single crystal substrate (SRO deposited by PLD) and from the second column are attributed to polycrystalline PZT film deposited by spin-coating on Pt-coated Si substrate.

**Figure 5 nanomaterials-11-02124-f005:**
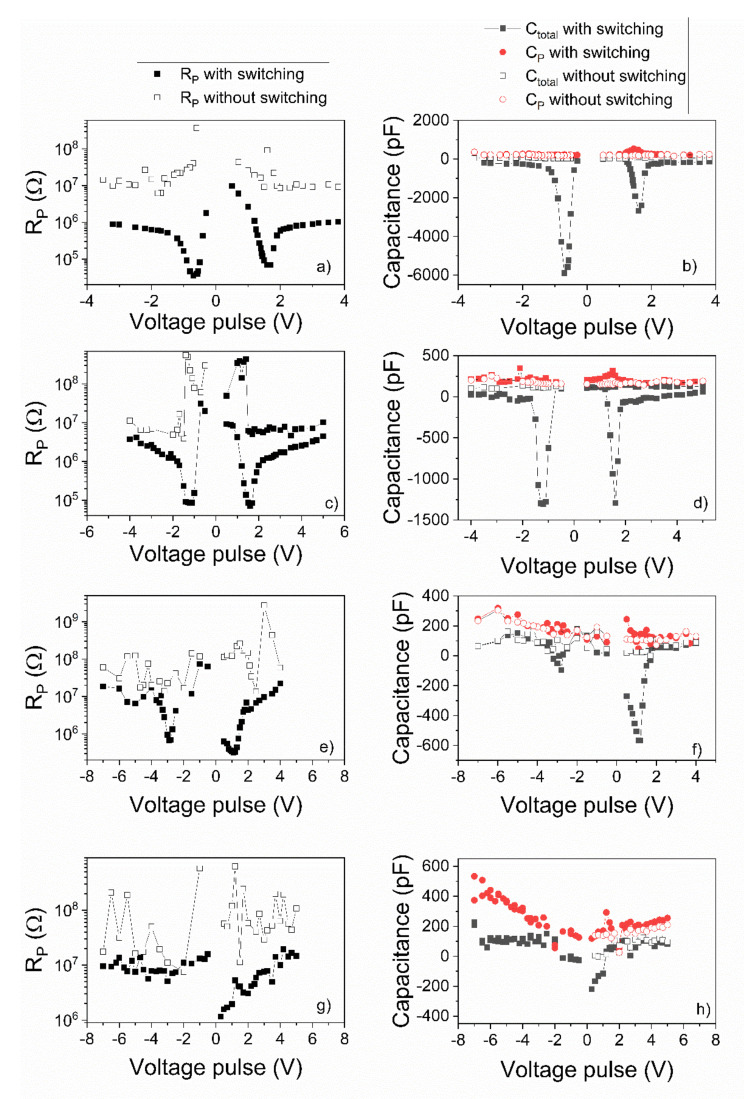
The dependence of the equivalent parallel resistance (R_P), total capacitance (C_total) and parallel capacitance (C_P) on the voltage pulse amplitudes. First (**a**,**b**) and second (**c**,**d**) lines are attributed to the epitaxial PZT thin films deposited from the commercial target and from “pure” target, respectively. The 3rd (**e**,**f**) and 4th (**g**,**h**) lines are attributed to the polycrystalline PZT film deposited by spin-coating on the SRO/STO single crystal substrate (SRO deposited by PLD) and on the Pt-coated Si substrate, respectively.

**Figure 6 nanomaterials-11-02124-f006:**
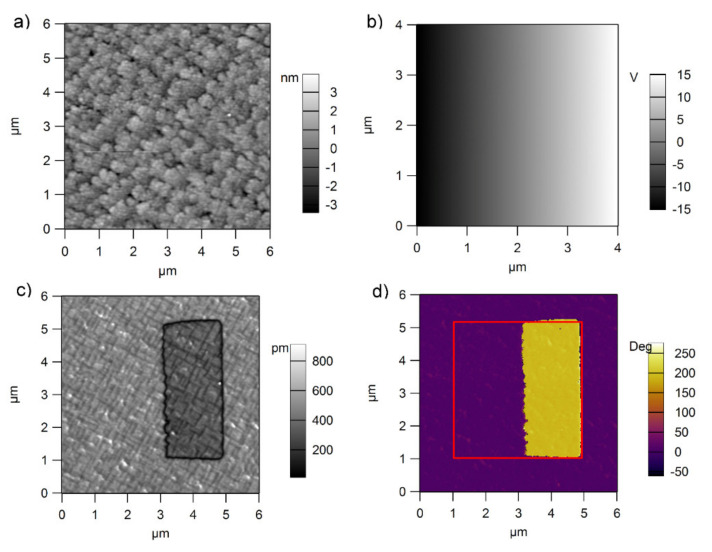
Topography (**a**), poling map (**b**) amplitude (**c**) and phase (**d**) of the PFM signal obtained after poling in the case of the PZT films deposited from commercial targets on a single crystal STO substrate with a bottom SRO electrode. The root mean square on the topography image (RMS) was estimated to about 0.5 nm.

**Figure 7 nanomaterials-11-02124-f007:**
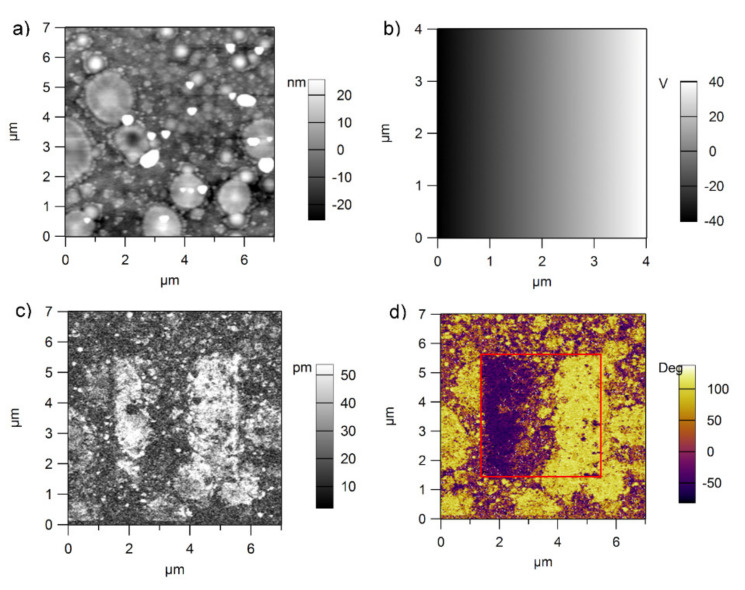
Topography (**a**), poling map (**b**) amplitude (**c**) and phase (**d**) of the PFM signal obtained after poling in the case of the PZT films deposited by sol-gel on an Si substrate with a bottom Pt electrode. The RMS was estimated to about 15 nm.

**Figure 8 nanomaterials-11-02124-f008:**
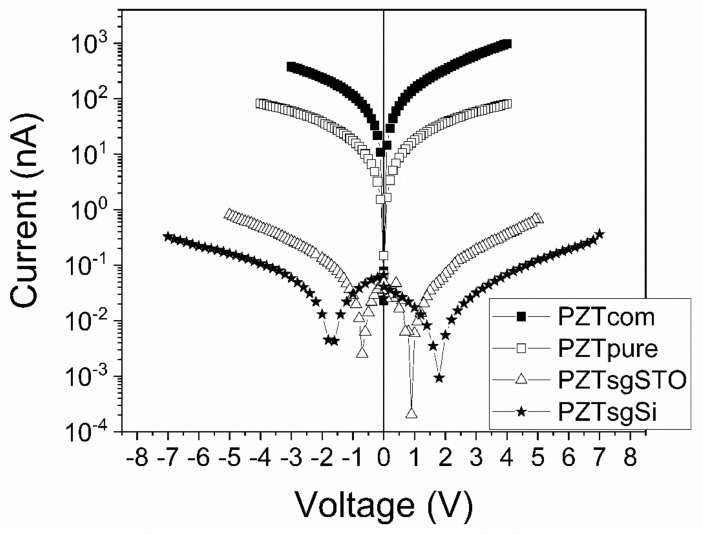
The I-V characteristics for the films investigated in this study: PZT deposited from commercial target (PZT com); PZT deposited from the in-house made, pure target (PZT pure); PZT deposited by sol-gel on STO substrate with bottom SRO electrode (PZT sgSTO); and PZT deposited by sol-gel on an Si substrate with a bottom Pt electrode (PZT sgSi).

**Figure 9 nanomaterials-11-02124-f009:**
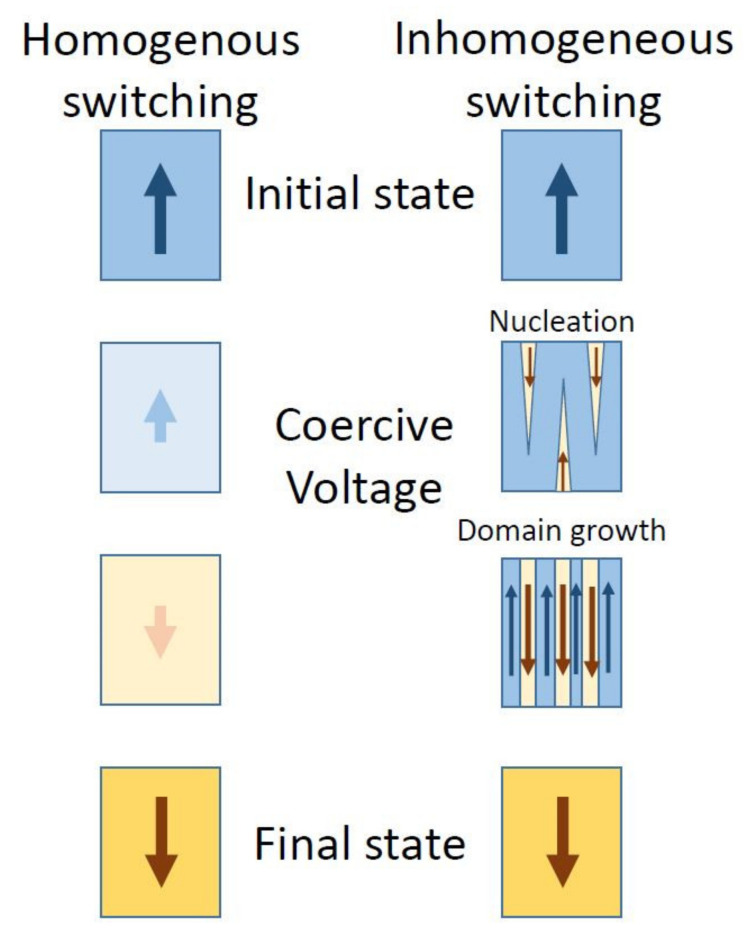
Left-homogeneous polarization switching, without domain formation (there is no nucleation phase of domains with opposite orientations of polarization; polarization merely flips from one direction to the other). Right-inhomogeneous switching, with domain formation (nuclei with opposite directions of polarization occur first, then they grow and coalesce until polarization is switched in the entire volume).

**Table 1 nanomaterials-11-02124-t001:** Values of remnant polarization Pr, coercive field Ec and internal electric field Ein for the PZT films of different structural qualities.

Sample	Pr (C/m^2^)	Ec (MV/m)	Ein (MV/m)
PZT—PLD grown from commercial target	0.87	16.25	4.37
PZT—PLD grown from pure target	0.65	17.95	2.65
PZT—sol-gel on SRO/STO substrate	0.295	16.5	−8
PZT—sol-gel on Pt/Si substrate	0.145	17.2	−2.8

## Data Availability

The data that support the findings of this study are available from the corresponding author upon reasonable request.

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
