# Peer review of "Homogeneous versus Inhomogeneous Polarization Switching in PZT Thin Films: Impact of the Structural Quality and Correlation to the Negative Capacitance Effect"

_nanomaterials, 2021, doi:10.3390/nano11082124_

Round 1

Reviewer 1 Report

The manuscript studies the polarization switching in different PZT films in depth and gives a novel polarization switching mechanism, i.e., the mechanism of homogeneous polarization switching in epitaxial PZT films. It is very good for understanding the complicated polarization switching either in films or body materials such as ceramics and single crystals. Some improvements are suggested as follows,

  1. The authors wrote that the concentration of the impurities in the targets was estimated by performing inductive coupled plasma-mass spectroscopy (ICP-MS) and X-ray fluorescence experiments. It was found that the amount of impurities acting as donors or acceptors is at least two times larger in the commercial target compared to pure one. Could you give the analysis results of their impurities?
  2. Could you introduce the measure devices used to study the electrical performance of films?
  3. Could you introduce the films dimension or their length and width?
  4. It’s suggested that, if possible, you might add a figure to illustrate the different mechanism related to homogeneous versus inhomogeneous polarization switching in PZT thin films. This figure may help readers to understand the polarization switching mechanism more clearly and may be very welcome by them.

Reviewer 2 Report

The manuscript is well organized and illustrates the scientific content of the work. This paper is publishable after some minor revisions.
Keywords must be re-ordered.

Page 2, lines 90-92. Give more details and explain this sentence.

Insert statistical analysis of the results.

Page 3, lines 96-99: provide the names of the used equipment and tools.  

Page 4, line 144: minor mistake: “....istructure...”

Page 6, line 209: minor mistake: “superscript [25].”

Page 8, line 230: Why it was chosen: “The dynamic dielectric characterization method”. Specify the advantages and disadvantages of this method. Are there any other methods?

Page 11, Figs. 6 and 7: Topography (a). Only if it is possible, could you specify some height parameters for surface texture, such as a) Height parameters: Root mean square height Sq [nm]; Skewness Ssk [-]; Kurtosis Sku [-]; Maximum peak height Sp [nm]; Maximum pit height Sv [nm]; Maximum height Sz [nm]; Arithmetic mean height Sa [nm]. b) Fractal parameter: Fractal dimension.

Page 12, line 332-334: insert the range of values for: “can be larger or lower”

If possible, I recommend to be cited the following reference:

1) F. M. Mwema, E. T. Akinlabi, O. P. Oladijo, O. S. Fatoba, S. A. Akinlabi, Ş. Ţălu, Advances in manufacturing analysis: fractal theory in modern manufacturing. In "Modern Manufacturing Processes", 1st edition, section 1, chapter 2, pages 13-39. DOI: 10.1016/B978-0-12-819496-6.00002-6. Edited by: K. Kumar, J. P. Davim. Woodhead Publishing Reviews: Mechanical Engineering Series, USA, 2020.

This paper presents an interesting approach and deserved to be published after the mentioned revisions.
